# A Precise Diagnosis Method of Structural Faults of Rotating Machinery based on Combination of Empirical Mode Decomposition, Sample Entropy, and Deep Belief Network

**DOI:** 10.3390/s19030591

**Published:** 2019-01-30

**Authors:** Zhaoyi Guan, Zhiqiang Liao, Ke Li, Peng Chen

**Affiliations:** 1Graduate School of Bioresources, Mie University, 1577 Kurimamachiya-cho, Tsu, Mie 514-8507, Japan; endlesswalts00@163.com (Z.G.); chen@bio.mie-u.ac.jp (P.C.); 2Jiangsu Key Laboratory of Advanced Food Manufacturing Equipment and Technology, Jiangnan University, Wuxi 214122, China

**Keywords:** rotating machinery, structural faults, symptom parameters, precise diagnosis, deep learning

## Abstract

To precisely diagnose the rotating machinery structural faults, especially structural faults under low rotating speeds, a novel scheme based on combination of empirical mode decomposition (EMD), sample entropy, and deep belief network (DBN) is proposed in this paper. EMD can decompose a signal into several intrinsic mode functions (IMFs) with different signal-to-noise ratios (SNRs) and sample entropy is performed to extract the signals that carry fault information with high SNR. The extracted fault signal is reconstructed into a new vibration signal that will carry abundant fault information. DBN has strong feature extraction and classification performance. It is suitably performed to build the diagnosis model based on the reconstructed signal. The effectiveness of the proposed method is validated by structural faults signal and the comparative experiments (BPNN, CNN, time-domain signal only, frequency-domain signal only). The results show that the diagnosis accuracy of the proposed method is between 99% and 100%, the BPNN is less than 25%, and the CNN is between 70% and 95%, which means the verified, proposed method has a superior performance to diagnose the structural fault.

## 1. Introduction

Rotating machinery covers extensive mechanical equipment and plays an important role in industrial production. It generally serves in a complicated and harsh environment, and may experience faults caused by the environment [1]. It is of important significance to guarantee safe and reliable running of large rotating machinery.

Structural faults are the most common fault in rotating machinery. It not only causes direct baneful influences on performance of equipment and quality of products, but also it causes excessive stress on surrounding components (such as bearings and gears) and causes secondary failure [2]. Therefore, early diagnosis of structural faults and diagnosis of fault types are extremely important subjects [3,4]. Structural faults of rotating machinery often include unbalance, misalignment, and looseness of fasteners, which are caused by machinery structural defects. These structural faults lead to a relatively low-frequency fault vibration. Compared with other types of faults, the faults caused by structural faults have two prominent common features, namely, component changes in the spectrum of the vibration signal at the rotating frequency and its harmonic frequency. In particular, vibration signals of different structural faults under low-rotating speed present extremely similar characteristics on the frequency spectra, which bring a great difficulty for extraction of fault characteristics and diagnosis of fault types. Although some studies have revealed part of spectral characteristics of a series anomalous state, it is still insufficient to accurately draw a distinguishing line between different structural faults [5].

It is a common method in rotating machinery fault diagnosis to analyze fault vibration signals and disclose fault characteristics by effective signal processing technology [6,7]. Whether key fault information can be extracted from vibration signals for diagnosis is an important challenge of signal processing technology. Traditional signal processing technologies covers analysis of time-domain signals and frequency-domain signals [8,9]. They perform well in the analysis of fault characteristics.

In frequency-domain analysis, characteristics of vibration signals in the frequency domain can be observed more easily by Fourier transform. In addition, explicit fault characteristics and diagnosis information can be acquired through frequency spectra of vibration signals [10]. However, signals of different types of fault caused by structural faults present extremely similar features on frequency spectra, especially under a low rotating speed [11]. This increases the difficulty to extract fault characteristics and diagnosis of fault types significantly. The traditional frequency-domain analysis methods are unable to realize precise diagnosis of structural faults of such rotating machinery.

Compared with other machine faults, structural faults are prominent for changes of rotating frequency and changes of harmonic component. Moreover, vibration signals of structural faults are dynamic nonlinear system signals. Traditional time-domain analyses are mainly based on the hypothesis that the process of signal production is static and linear. Hence, they may make the wrong judgment if fault signals are dynamic nonlinear systematic signals caused by structural faults [12,13]. Some studies introduced empirical mode decomposition (EMD) in order to process these dynamic nonlinear systematic signals in the rotating machinery fault. As a strong time-frequency analysis technology, EMD is an adaptive signal processing technology applicable to a nonlinear and non-static process [14]. It has been widely used in many fields, including injection control, modeling, speech recognition, and system control. In many studies on EMD, EMD has been applied in fault diagnosis of rolling bearing, gears, and rotors [15]. Nevertheless, these studies basically focus on one component of machineries rather than the type of fault [16]. It is necessary to emphasize on integrity of the equipment, but few have discussed the precise diagnosis method of rotating machinery structural faults. In addition, the original EMD method involves many problems, such as mode mixing, end effect, interpolation problem, end standards, and optimal intrinsic mode function (IMF) selection [17]. These may make EMD produce meaningless or unnecessary intrinsic mode functions (IMFs), which may decrease the fault diagnosis accuracy and even misguide the diagnosis decision. It is necessary to further improve EMD or combine it with other processing techniques.

With the popularization of the artificial neural network, the fault diagnosis becomes more and more intelligent and accurate under the assistance of neural networks by its strong pattern recognition capability. Even though some studies have implemented a neural network-based fault diagnosis, the types of faults they can diagnose are specific fault states and are very limited [18]. We have not found a study that uses the neural network to precisely diagnose structural faults, especially at low rotating speeds. Moreover, some studies focus on fault diagnosis of single part, instead of the global precise diagnosis of multiple fault types of multiple parts [19]. Other conventional discriminant methods like support vector machine (SVM) have evident advantages in binary classification problems. However, SVM still fails to achieve precise diagnosis of structural faults, which is related with the difficult extraction of fault characteristics. 

Deep learning possesses considerable advantages in pattern recognition and it has been highly used in many fields [20,21,22,23]. It makes the precise fault diagnosis based on deep learning, especially precise diagnosis of structural faults, possible. Appropriate processing of diagnosis signals before deep learning can further increase the diagnosis accuracy of machine faults [24]. In this study, diagnosis accuracy is too far from precise diagnosis when the original time-domain signals are input into the deep belief neural network (DBN). Although the diagnosis accuracy is increased significantly by inputting the frequency-domain signals, the diagnosis accuracy under low rotating speed is still insufficient to meet the requirements of a precise diagnosis. In order to increase the accuracy of a precise diagnosis, the fault information extraction method that combines EMD and sample entropy was proposed. The original signal is decomposed into many intrinsic mode functions with different frequency domains, and the sample entropy is calculated for extracting the signals that carry fault information with high signal-to-noise ratio (SNR). The extracted fault signal is reconstructed into a new vibration signal that will carry abundant fault information. The reconstructed signal treated the input and brought satisfying diagnosis accuracy under different rotating speeds, which meets the requirements of precise diagnosis. Structural faults of rotating machinery could be recognized by this method. On this basis, an intelligent diagnosis system for structural faults could be constructed.

Procedure of the proposed method is shown in Figure 1 and includes the following steps:measuring vibration signals of the targeted rotating machinery (hereinafter referred to as the diagnosed signals);decomposing the original vibration signal into several IMFs by EMD;calculating sample entropy of IMFs and screen the signals that contain many fault information;reconstructing the screened signals into new characteristic signals;implementing FFT to new characteristic signals, and then gain their frequency-domain characteristic signals;inputting the frequency-domain characteristic signals into the trained DBN and get the final diagnosis results.

The rest part of this paper is organized as follows. Section 2 introduces the theory of the proposed method including EMD, sample entropy, extracted fault signal reconstruction, and the DBN diagnosis model. In Section 3, the effectiveness of the proposed method is validated by structural faults signal under different rotating speeds. Section 4 presents a comparison with the BPNN model, the CNN model, the time domain signal only, and the frequency domain signal only. Lastly, Section 5 summarizes the conclusions.

## 2. Basic Principle of the Proposed Method

### 2.1. Extraction of Fault Information

Fault information in diagnosed signals shall be extracted before the training and use of the DBN in order to increase diagnosis accuracy based on DBN. This can be realized by combining EMD and sample entropy.

#### 2.1.1. EMD

EMD is an adaptive signal analysis and processing technique. In fact, it makes stationary processing of data sequences or signals. EMD has evident advantages in processing non-stationary and nonlinear data and has high SNR. Since vibration signals of different structural faults often have many similar characteristics on the waveform and frequency spectra, it is difficult to distinguish different types of structural faults. To extract fault information and improve the diagnosis accuracy, the diagnosed signal was decomposed into several IMFs by EMD.

Instantaneous frequency at any one point of IMFs is meaningful. At any moment, signals can contain several IMFs. Composite signals are formed when IMFs overlap mutually.

The decomposition process is introduced as follows.

Effective EMD process of one given signal *x*(*t*) contains the following steps:

(1) Find out all extreme points of the original signal *x*(*t*), including the maximum point (*xmax*(*t*)) and the minimum point (*xmin*(*t*)).

(2) Form the lower envelope *eminx*(*t*) of the minimum point and the upper envelope of maximum point *emax*(*t*) by using the interpolation method.

(3) Calculate the mean:*m*(*t*) = (*eminx*(*t*) + *emax*(*t*))/2(1)

(4) Draw the details:*d*(*t*) = *x*(*t*) − *m*(*t*)(2)

(5) If *d*(*t*) has a negative local maximum and a positive local minimum, it is not an IMF and screening shall be continued. In other words, all extreme points of *d*(*t*) shall be recognized and all above steps shall be repeated until meeting the given standards. If the screening process is accomplished successfully, the first IMF is extracted and it shall be subtracted from the original data. Next, repeat this process to get the next IMF until all IMFs are extracted.

#### 2.1.2. Sample Entropy

Each diagnosed signal can decompose several IMFs by the above EMD. However, only several signals that contain fault information have to be screened and reconstructed into new vibration signals for a more precise diagnosis. This was accomplished by sample entropy in this study. 

Sample entropy (SampEn) is an improved method to measure the complexity of time sequence based on approximate entropy (ApEn). It has been used in assessment of complexity of physiological time series and diagnosis of the pathological state. In this study, sample entropy can be used as the standard to judge how much fault information each signal contains.

The calculation process of sample entropy is introduced as follows:

The original data is set as a time series with a length of *N*, which is expressed as {u(i):1≤i≤N}.

(1) Construct a group of m-dimensional vectors: *X*(1), *X*(2), …, *X*(*N* − *m* + 1), and where *X*(*i*) = {*u*(*i*), *u*(*i* + 1), …, *u*(*i* + *m*)}.

(2) The distance *d[X(i),X(j)]* between vectors *X*(*i*) and *X*(*j*) as the one with the maximum difference in corresponding elements is: (3)$$d\left[X(i),X(j)\right]=\underset{k=0\sim m-1}{\max}\left|u(i+k)-u(j+k)\right|$$

(3) For each {*i*:1 ≤ *i* ≤ *N* − *m* + 1}, a statistics on number of *d*[*X*(*i*),*X*(*j*)] < *r* is carried out when the allowable deviation is r, which is recorded as *Nm*(*i*). Meanwhile, the ratio between *Nm*(*i*) and the total distance is calculated, which is denoted as the equation below. 

(4)$$C_{i}^{m}\left(r\right)=N_{m}\left(i\right)/\left(N-m\right)$$

(4) Calculate the mean of all *i*, which is denoted as the formula below. 

(5)$$\phi^{m}\left(r\right)=1/\left(N-m\right)\sum_{i=1}^{N-m}C_{i}^{m}\left(r\right)$$

(5) Increase the dimension m by *1* to *m* + 1. Repeat Steps (1)–(4) to get:(6)$$C_{i}^{m+1}\left(r\right)=N_{m+1}\left(i\right)/\left(N-m+1\right)$$

(7)$$\phi^{m+1}\left(r\right)=1/\left(N-m+1\right)\sum_{i=1}^{N-(m+1)}C_{i}^{m+1}\left(r\right)$$

(6) Theoretically, the sample entropy *SampEn*(*N*,*m*,*r*) of this series is shown below.

(8)$$SampEn\left(m,r\right)=\underset{N\to \infty }{\lim}\left\{-\ln\left[\phi^{m+1}\left(r\right)/\phi^{m}\left(r\right)\right]\right\}$$

Practically, *N* cannot be ∞. When *N* is a limited value, it is estimated that:(9)$$SampEn\left(N,m,r\right)=-\ln\left[\phi^{m+1}\left(r\right)/\phi^{m}\left(r\right)\right]$$

Since sample entropy is an index that mainly assesses the complexity of the signal, it is inadequate for extracting essential fault information for precise diagnosis. For effective extraction of fault information, the sample entropies of different IMFs, which are decomposed by the EMD method from both the diagnosed signal and the normal state signal, have to be calculated. The ratio between IMFs of diagnosed signal and IMFs of the normal state signals was used as the reference to screen the signal containing many fault information.

#### 2.1.3. The Reconstruction of Extracted Fault Signal

Through the empirical modal decomposition above, the original vibration signal is decomposed into several intrinsic mode functions. Each intrinsic mode function contains varying degrees of fault information. The sample entropy can screen IMFs that contain fault information. New vibration signals can be reconstructed by integrating these IMFs. The new signals have higher SNR than the original vibration signal and can increase the diagnosis accuracy effectively. In this study, the first three IMFs with the highest sample entropy ratios (farther from 1) were chosen from each group to reconstruct the new signals. New signals were used to replace the original vibration signal in a precise diagnosis. The extraction of fault information and the reconstruction of the vibration signal are shown in Figure 2. The reconstructed signals are shown in Figure 3.

### 2.2. Structure and Principle of DBN

DBN is a probability generation model and a multi-hidden layer neural network formed by multiple restricted Boltzmann machines (RMB). The DBN model can extract features layer by layer from the original data through layered piling of RBM, which shows high-level expressions. The core of DBN is to optimize connection weight of the deep neural network by using the layer greedy learning algorithm [25,26]. First, characteristics in data shall be mined effectively by unsupervised layer-wise training. Second, the classification ability of DBN shall be optimized by the reverse supervised fine adjustment based on adding the corresponding classifiers.

DBNs are composed of multiple RMBs. The structure of one DBN is shown in Figure 4. This is “restricted” into one visible layer and multiple hidden layers. There are connections between layers, but there is no connection between units in each layer [27]. Units in hidden layers were trained to capture characteristics of input data in the visible layer. RBM is a neural sensor composed of one explicit layer and one hidden layer. Neurons between the explicit layer and the hidden layer adopt two-way full connections. Three RBMs were “connected in series” from the bottom upwards, so the DBN in Figure 3 was constructed. The hidden layer of the first RBM is the explicit layer of the second RBM, and the output of the second RBM is the input of the third one. During training, RBM of the current layer can only be trained after the previous layer of RMB has been trained completely. This shall be implemented until the last layer. The structure of RBM is shown in Figure 5.

To train the DBN, unsupervised pre-training based on RBM was performed first. Weights were initialized by the comparison divergence algorithm, which was followed by supervised optimized training. During supervised optimized training, it has to get a certain output from the input by using the forward propagation algorithm. Next, weights and offsets of the network shall be updated by the backward propagation algorithm. The training process of DBN is shown as follows. When training DBN, a layer-by-layer unsupervised method is used to learn the parameters. As shown in Figure 6, the data vector *x* and the first hidden layer are treated as one RBM, and the parameters of the RBM (the weights of *x* and *h_1_*, the offset of each node of *x* and *h_1_*, etc.) are trained, and then fixed. By treating *h_1_* as the visible vector, *h_2_* as the hidden vector, and training the second RBM, the parameters of RBM can be obtained and then fixed. Afterward, the RBM composed of *h_2_* and *h_3_* is trained.

EMD was performed to experimental data under different states (labels). Signals containing abundant fault information were screened, according to sample entropy ratio and then reconstructed into new vibration signals. Frequency spectra and labels of these reconstructed signals were used to train the neural network. The training for the proposed model is shown in Figure 7.

## 3. Precise Diagnosis of Structural Faults of Rotating Machinery

### 3.1. Diagnosis Process and Content of the Verification Experiment

In this chapter, the proposed method was used to precise diagnosis of structural faults of rotating machinery. First, vibration signals of different structural faults shall be collected (Figure 8). The following procedure was repeated during the collection of vibration signals under each state.

The rotating machinery simulator in Figure 9 was used to reproduce structural faults of rotating machinery. The layout of acceleration sensors is shown in Figure 10. They measured the vibration acceleration signals on two bearing supports along five directions (left vertical direction, right vertical direction, left horizontal direction, right horizontal direction, and axial direction).

It can be seen from Figure 11 that the preset structural faults include the misalignment state (angular misalignment and offset misalignment), the unbalanced state, and the looseness state (loss of pedestal and bearing pedestal). The misalignment state can be set by adjusting the position and angle of the spindle through the assistant positioning of the shaft laser alignment meter. The unbalance state was set by configuring different weights through the flange plate. The looseness state could be set by adjusting the tightness of bolts on the bearing box and pedestal.

### 3.2. Extraction Results of Characteristic Signals

In this study, the measured rotating speeds are 300, 500, 700, and 900 rpms, respectively. Acceleration sensors are used to measure the vibration signal in five directions. The vibration signal is stored by the data collector and converted into the csv format for subsequent processing. The sampling frequency is 10,000 Hz, the sampling time is 7 s, and the data length is 70,000. The original vibration signals and the EMD results are shown in Figure 12, Figure 13, Figure 14 and Figure 15.

It generally can be concluded from EMD results that the first IMFs among 10 decomposed IMFs have relatively dense images and may contain abundant fault characteristics. The rest of the IMFs have relatively sparse images and may contain few fault characteristics. However, this was inadequate to screen signals containing abundant fault information. To screen signals containing many fault information, sample entropy of each IMF shall be calculated.

Parameters were set as follows before calculating the sample entropy: the dimension m and threshold r of the constructed samples were m = 2500 and r = 0.008, respectively. Due to the size of mass data in the original signal, it took a considerable long time to calculate sample entropy. In this study, 5000 sample points in the original signal were selected to calculate sample entropy, which could increase the calculation efficiency and shorten calculation time significantly. The ratio between sample entropy of IMFs of different structural faults and the sample entropy of IMFs of normal state can be expressed in Table 1, Table 2, Table 3 and Table 4.

In each group of IMFs, the first three IMFs with the highest ratio of sample entropy were used to reconstruct the new vibration signals, which are shown in Figure 16 below.

Compared with the original vibration signal, the reconstructed vibration signals had fewer impacts from noises in view of images, and a higher proportion of fault information. Diagnosis based on reconstructed vibration signals can further increase the accuracy of precise diagnosis. 

### 3.3. Diagnosis Results Based on DBN

#### Comparison of Diagnosis Results Based on Time-Domain Signals, Frequency-Domain Signals, Combination of Frequency-Domain Signals, and Fault Information Extraction

To verify the validity of the proposed method, diagnosis results based on time-domain signal, diagnosis results based on frequency-domain signal, and diagnosis results based on the combination of frequency-domain signals, and the fault information extraction method were compared.

First, parameters of the neural network were set uniformly.

DBN size was set 100–50 (the explicit layer had 100 neurons and the hidden layer had 50 neurons).The number of iteration for samples was 100.The batch size was 160.Learning rate was 0.01.

When only time-domain signals were used in the diagnosis, the following occurred.

Table 5 showed that the diagnosis accuracy was very low under low-rotating speeds and it was difficult to realize precise diagnosis of structural faults when the diagnosed signals were time-domain signals. Based on the above analysis, structural faults have prominent characteristics, that is, changes of the rotating frequency and its harmonic component. This indicates that fault characteristics were mainly reflected on frequency-domain signals. As a result, it is difficult to extract and learn fault characteristics when using time-domain signals only in the diagnosis. The diagnosis accuracy is low accordingly.

When only frequency-domain signals were used in diagnosis, Table 6 occurred.

When the diagnostic signal was a frequency domain signal, the diagnostic accuracy is very high at high rotating speeds (700 rpm and 900 rpm), but the accuracy of the precision diagnosis is very low at low rotating speeds (300 rpm and 500 rpm). According to above analysis, the most prominent common features of structural faults are changes of rotating frequency and its harmonic components. Different types of structural faults have very similar characteristics, especially under a low rotating speed. Therefore, the diagnosis accuracy based on frequency-domain signals was not satisfying under low rotating speed despite the overall diagnosis accuracy being improved significantly. Fault characteristics have to be further extracted and SNR of signals need to be increased in order to further improve the diagnosis accuracy.

After the decomposition of the original signal by EMD and the selection of intrinsic mode function with high signal-to-noise ratio based on the sample entropy, the selected intrinsic mode functions are reconstructed into new vibration signals. The number of samples based on the reconstructed signals for each state is 100, the length of each sample is 1000, 80% of the total number of samples are randomly sampled for training, and the rest of the samples are used to verify the trained model. There are four types of states in this experiment (normal state, misalignment state, unbalance state, and looseness of fasteners), so the total number of training samples is 320, and the total number of test samples is 80. The structure of the deep belief neural network is [1000,100,50,4], the number of iterations of the sample is 100, and the learning rate is 0.01. Based on the trained deep belief neural network, the diagnostic accuracy of the test samples is shown in Table 7 below.

During EMD processing of diagnosed signals, IMFs that contain abundant fault information were chosen according to the ratio with sample entropy of normal state and they were reconstructed into new vibration signals. The frequency-domain signals of these new vibration signals were used in diagnosis, which achieved very high accuracy under both high and low rotating speeds. The precise diagnosis was realized. 

In the comparison experiment, the method of using the time domain signals only, the method of using frequency domain signals only, and the proposed method are performed. The fault characteristics in the time domain signal is difficult to express, which will result in low diagnostic accuracy when only using time domain signals for diagnosis. When only using frequency domain signals for diagnosis, although the diagnostic accuracy is greatly improved compared with the method of only using time domain signals, the faults of the structural faults often have similar characteristics and the diagnostic accuracy is not satisfied. In this paper, the method of extraction of fault information based on empirical mode decomposition and sample entropy is proposed to screen and reconstruct the vibration signals with high SNR for precision diagnosis, and the frequency domain signals of reconstructed signals are combined with a deep belief neural network, which has strong classification performance. This method can increase diagnosis accuracy significantly and realize a precise diagnosis of structural faults. This is because reconstructed vibration signals not only extract fault characteristics in original signals effectively, but also have extremely high SNR, which is conducive for increasing diagnosis accuracy greatly. The diagnosis accuracy was verified by changing the parameters of DBN.
DBN size was set 100–75 (the explicit layer had 100 neurons and the hidden layer had 75 neurons).The number of iteration for the samples was 50.The batch size was 160.The learning rate was 0.05.Diagnosis results.
The diagnosis results show in Table 8.
DBN size was set to 100–100 (the explicit layer had 100 neurons and the hidden layer had 75 neurons).The number of iteration for the samples was 50.The batch size was 160.The learning rate was 0.02.Diagnosis results.
The diagnosis results show in Table 9.

It can be known from the above diagnosis results that the proposed method still maintains extremely high diagnosis accuracy after parameters of DBN are changed. This method can adjust parameter settings according to different needs and realize a precise diagnosis.

## 4. The Comparison with Diagnostic Results of Diagnostic Methods Based on Traditional Neural Network and Dimensionless Symptom Parameter

### 4.1. Common Symptom Parameters

In order to compare the diagnostic accuracy of the precision diagnostic method with the diagnostic methods based on other neural networks, it is necessary to extract the fault characteristics for precise diagnosis before using the traditional neural network for diagnosis. Dimensional and dimensionless symptom parameters are capable of extracting the characteristics of vibration signals and are commonly used for mechanical condition monitoring. These parameters allow for more sensitive detection of mechanical faults and allow for distinguishing fault types [28]. However, how to extract symptom parameters and whether the extracted symptom parameters are suitable for diagnosis has always been an important issue. In this study, the reference is made to some of the symptom parameters proposed in the relevant research, as well as the symptom parameters commonly used in statistics, which will be used in the diagnostic method based on a traditional neural network [29,30].

### 4.2. Diagnostic Method Based on the Back Propagation Neural Network

Back propagation neural network (BPNN) is the most common kind of neural network. It is a multi-layer feedforward network trained by error back propagation. The algorithm is called BP algorithm. The basic idea is the gradient descent method. Gradient search technology is used to minimize the error mean square error between the actual output value of the network and the expected output value [31]. In this study, the symptom parameters and their principal components are used as inputs to the BP neural network, respectively.

### 4.3. Diagnostic Method Based on the Convolutional Neural Network

The convolutional neural network (CNN) is a multi-layer feedforward neural network. It is good at dealing with related machine learning problems of images, especially large images [32]. It has excellent performance for large image processing and is widely used in image recognition. Through a series of methods, the convolutional network has successfully reduced the image recognition problem with huge data volume and enabled it to be trained. It includes convolutional layers and pooling layers. 

In this study, the symptom parameters and their principal components are reconstructed into a matrix of image forms and then used as input to a convolutional neural network. When the symptom parameters are used as input, each state signal is divided into 12 segments, and a set of special feature parameters can be calculated, according to each segmentation signal. The number of each group of feature parameters is 12. The symptom parameters are rearranged and constructed into a 12×12 new pixel matrix as an input to the CNN neural network.

### 4.4. The Summary of Results of Each Method Based on Neural Networks

To further verify the effectiveness of the proposed method, the results of the DBN network are compared with those of other neural networks (such as BPNN and CNN). The experimental data used are consistent with Section 3, and the diagnostic results are shown in Table 10.

From the results in Table 10, it can be seen that the accuracy of the diagnosis method based on BPNN is less than 25%, the accuracy of the diagnosis method based on CNN is between 70% and 95%, the accuracy of the diagnosis result using the time domain signal based on the DBN method is less than 25%, and the accuracy of the diagnostic results only using the frequency domain signal based on the DBN method is between 80% and 96%. These results show that the diagnostic accuracy of these methods is not as accurate as the method proposed in this paper.

## 5. Conclusions

In this study, to realize accurate detection and recognition of structural faults of rotating machinery, a precise diagnosis method based on fault information extraction that combines EMD and sample entropy as well as DBN was proposed. Diagnosis results based on time-domain signals, diagnosis results based on frequency-domain signals, and diagnosis results based on the combination of frequency-domain signals and fault information were compared. It finds that diagnosis accuracy based on time-domain signals was unsatisfying because time-domain signals are difficult to express all features of signals. Therefore, the time-domain signals fail to detect and recognize structural faults. The most prominent common features of structural faults are changes of rotating frequency and its harmonic components. Therefore, diagnosis accuracy based on frequency-domain signals was increased significantly compared with that based on time-domain signals. Since different types of structural faults have similar features in the frequency domain, the diagnosis accuracy based on frequency-domain signals under low rotating speed is unsatisfying and has to be improved. When fault information was extracted by combining EMD and sample entropy, the original vibration signals were decomposed and signals with high SNR that contain abundant fault characteristics were selected and reconstructed into new vibration signals. Frequency-domain signals of these new vibration signals were used in diagnosis, which achieves high accuracy under both high and low rotating speeds. In other words, the proposed method can detect and recognize structural faults under different rotating speeds. It realizes precise diagnosis of structural faults. Moreover, an intelligent precise diagnosis system can be constructed based on the proposed method. This is conducive to realize the goal of early diagnosis of structural faults in practical production. Compared with BPNN and CNN, the method proposed in this study was superior to other methods in the precision diagnosis of structural faults in rotating machinery, and has the highest diagnostic accuracy.

## Figures and Tables

**Figure 1 sensors-19-00591-f001:**
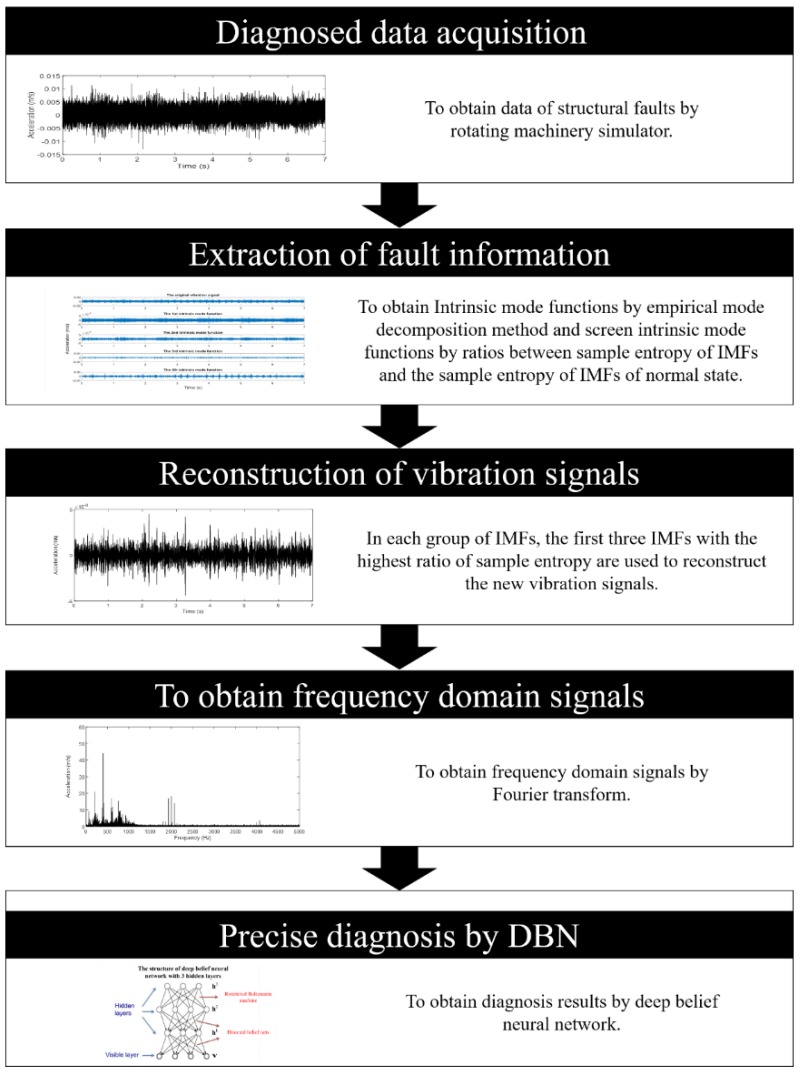
Precision diagnostic process based on deep belief neural networks.

**Figure 2 sensors-19-00591-f002:**
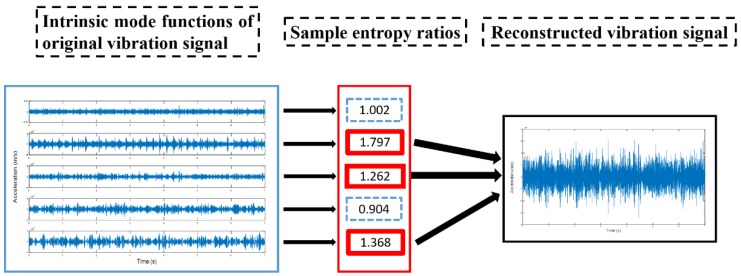
The process of extraction of fault information and reconstruction of extracted fault signal.

**Figure 3 sensors-19-00591-f003:**
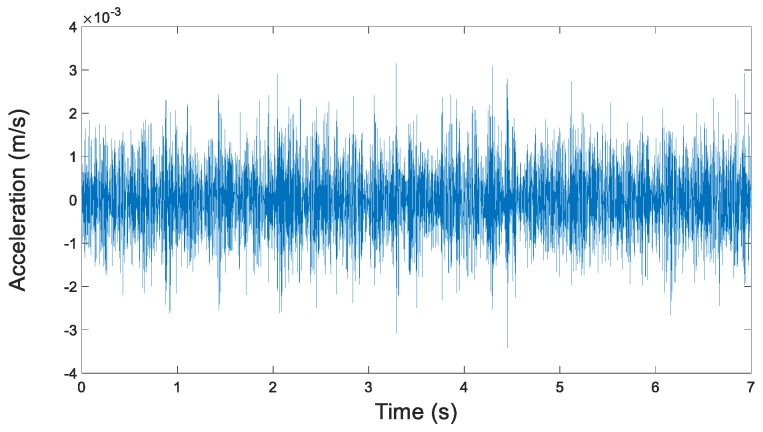
Reconstructed vibration signal (300 rpm, unbalance state).

**Figure 4 sensors-19-00591-f004:**
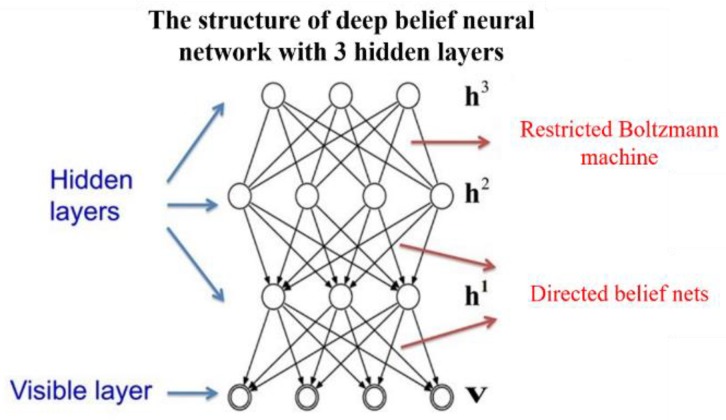
Structure of the deep belief neural network with three hidden layers.

**Figure 5 sensors-19-00591-f005:**
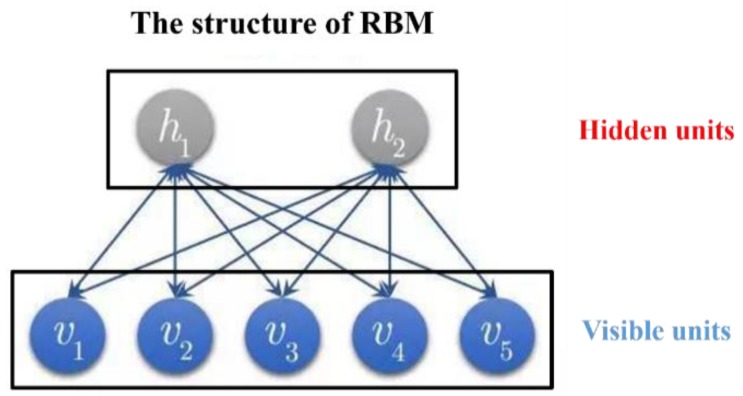
The structure of RBM (Restricted Boltzmann Machine).

**Figure 6 sensors-19-00591-f006:**
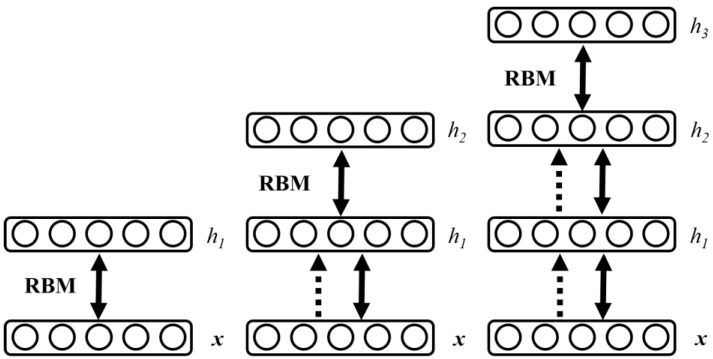
The training process for the Deep Belief Network (DBN).

**Figure 7 sensors-19-00591-f007:**
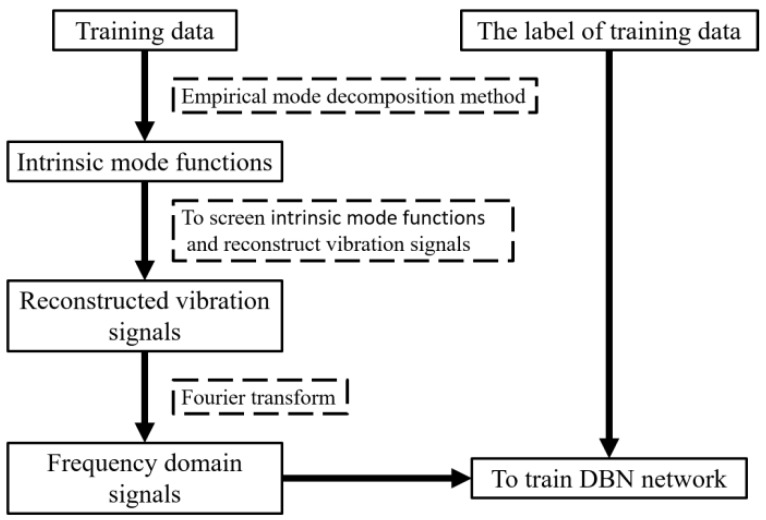
The training for the proposed model.

**Figure 8 sensors-19-00591-f008:**
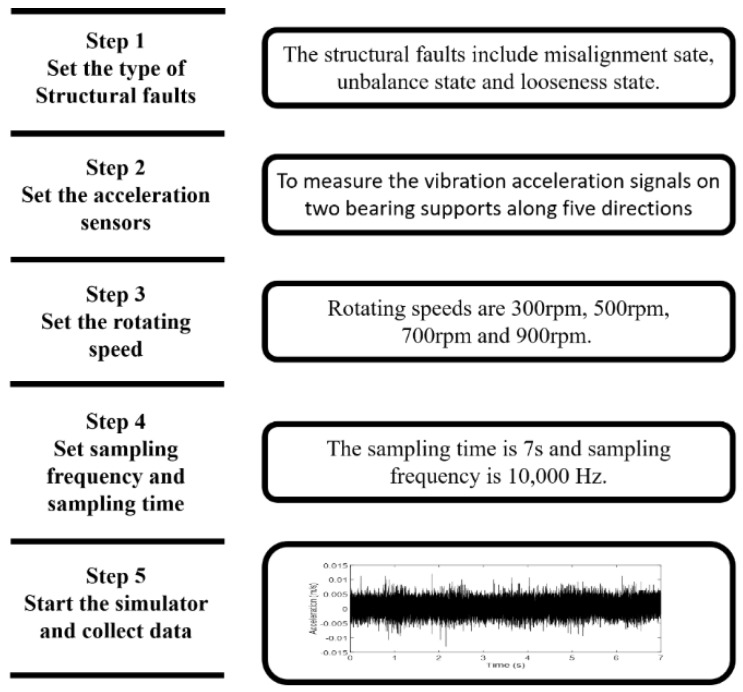
The process of signal collection.

**Figure 9 sensors-19-00591-f009:**
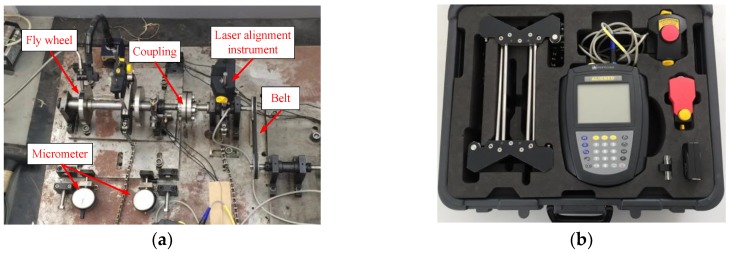
Rotating machinery simulator and laser alignment instrument. (**a**) Rotating machinery simulator. (**b**) Laser alignment instrument.

**Figure 10 sensors-19-00591-f010:**
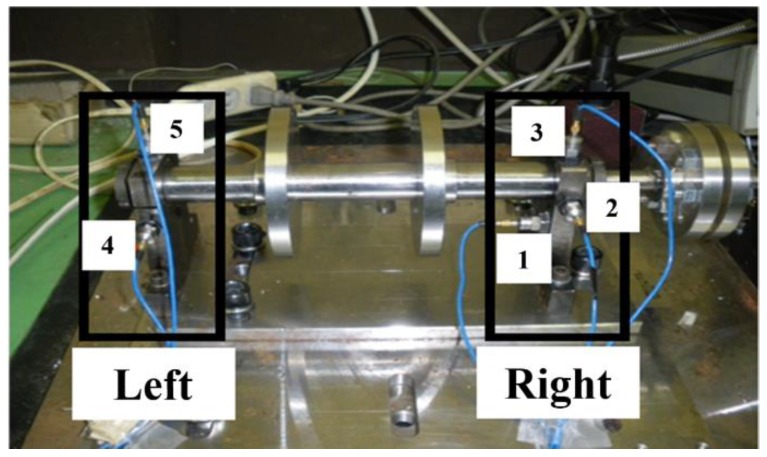
The layout of acceleration sensors.

**Figure 11 sensors-19-00591-f011:**
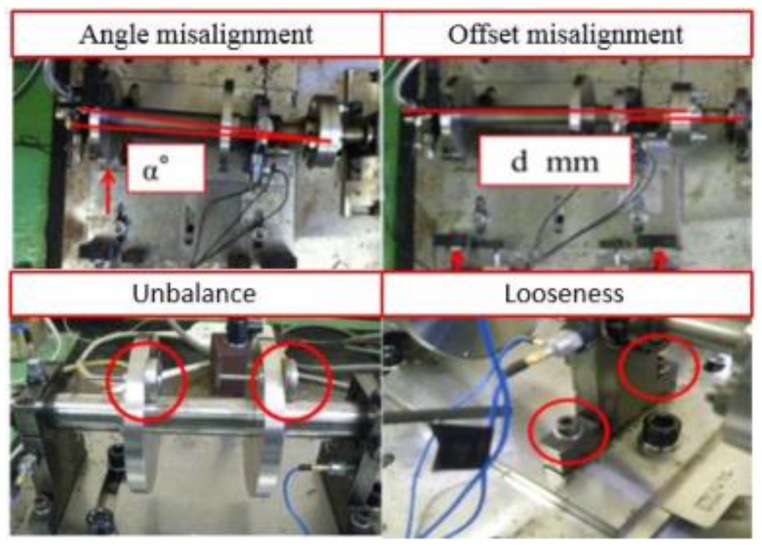
Set each structure faults status on the rotating machinery simulator.

**Figure 12 sensors-19-00591-f012:**
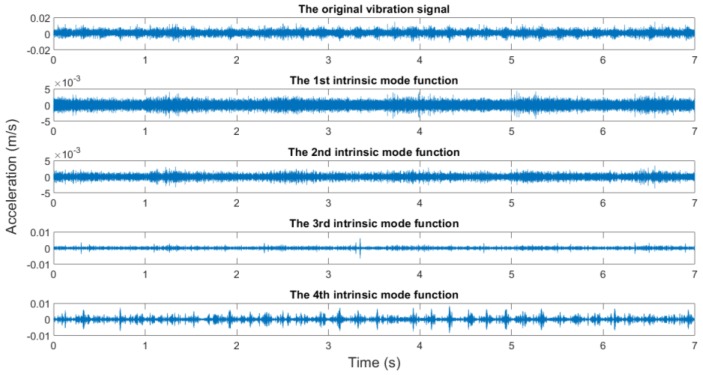
The original vibration signal and the EMD results (300 rpm, the looseness state of the pedestal).

**Figure 13 sensors-19-00591-f013:**
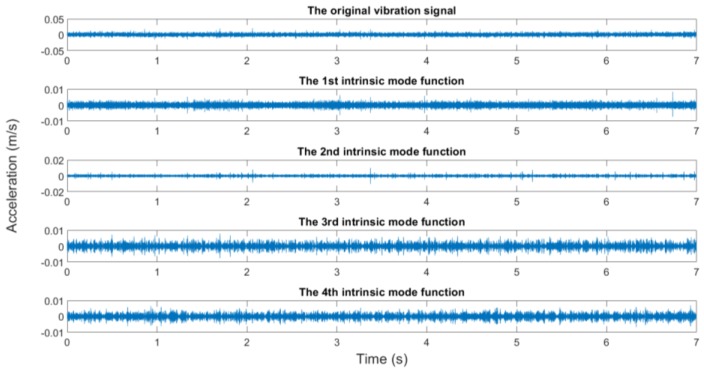
The original vibration signal and the EMD results (500 rpm, the looseness state of the bearing housing).

**Figure 14 sensors-19-00591-f014:**
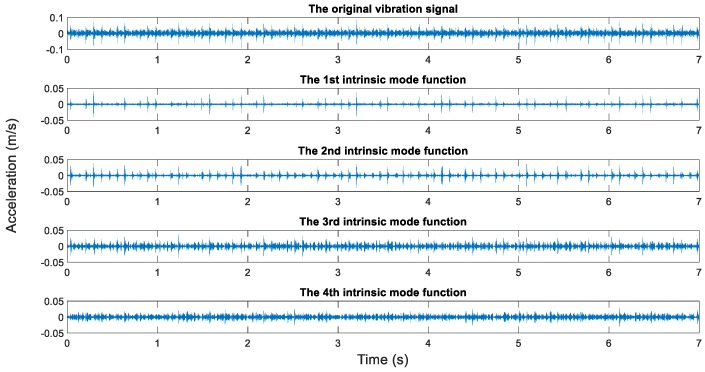
The original vibration signal and the EMD results (700 rpm, unbalance state).

**Figure 15 sensors-19-00591-f015:**
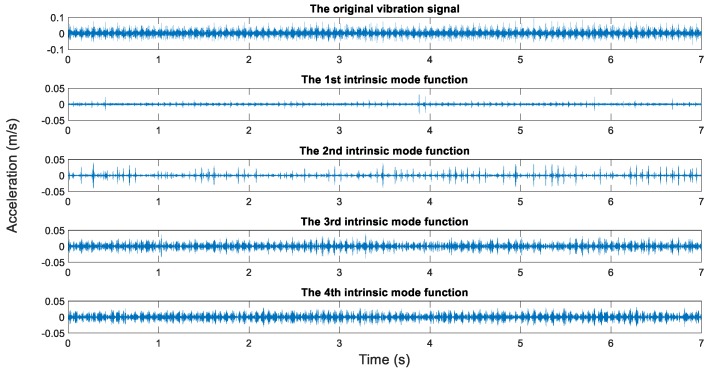
The original vibration signal and the EMD results (900 rpm, misalignment state).

**Figure 16 sensors-19-00591-f016:**
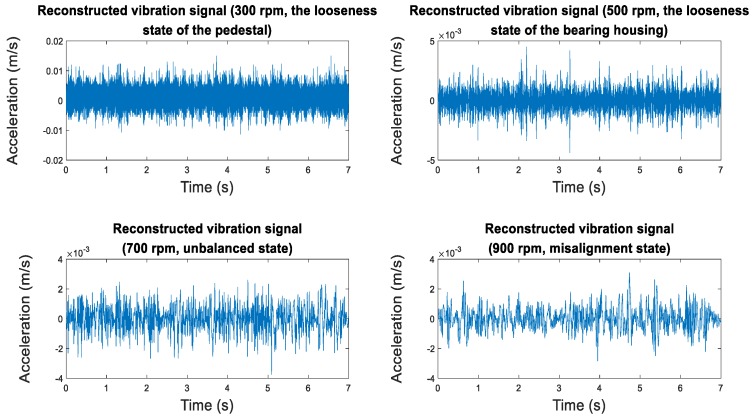
Reconstructed vibration signals.

**Table 1 sensors-19-00591-t001:** Ratios between sample entropy of IMFs and the sample entropy of IMFs of normal state (300 rpm, the looseness state of the pedestal).

The Numbering of the Intrinsic Mode Function	The Ratio with the Sample Entropy of the Reference State
1	1.006
2	1.006
3	1.006
4	1.003
5	0.380
6	1.314
7	1.248
8	1.004
9	1.004
10	1.004

**Table 2 sensors-19-00591-t002:** Ratios between sample entropy of IMFs and the sample entropy of IMFs of normal state (500 rpm, the looseness state of the bearing housing).

The Numbering of the Intrinsic Mode Function	The Ratio with the Sample Entropy of the Reference State
1	1.006
2	1.006
3	1.006
4	1.159
5	0.766
6	1.900
7	0.484
8	1.516
9	1.004
10	1.004

**Table 3 sensors-19-00591-t003:** Ratios between sample entropy of IMFs and the sample entropy of IMFs of a normal state (700 rpm, unbalanced state).

The Numbering of the Intrinsic Mode Function	The Ratio with the Sample Entropy of the Reference State
1	1.006
2	1.006
3	1.006
4	1.006
5	0.376
6	0.642
7	0.866
8	0.964
9	1.442
10	0.741

**Table 4 sensors-19-00591-t004:** Ratios between sample entropy of IMFs and the sample entropy of IMFs of normal state (900 rpm, misalignment state).

The Numbering of the Intrinsic Mode Function	The Ratio with the Sample Entropy of the Reference State
1	1.006
2	1.006
3	1.006
4	1.006
5	0.400
6	1.973
7	0.985
8	0.690
9	1.326
10	0.778

**Table 5 sensors-19-00591-t005:** Diagnostic results using time domain signals.

The Type of Diagnostic Data	Rotating Speed	Diagnostic Accuracy
Time domain signal	300 rpm	20%
Time domain signal	500 rpm	20%
Time domain signal	700 rpm	20%
Time domain signal	900 rpm	20%

**Table 6 sensors-19-00591-t006:** Diagnostic results using frequency domain signals.

The Type of Diagnostic Data	Rotating Speed	Diagnostic Accuracy
Frequency domain signal	300 rpm	80%
Frequency domain signal	500 rpm	82%
Frequency domain signal	700 rpm	98%
Frequency domain signal	900 rpm	96%

**Table 7 sensors-19-00591-t007:** Diagnostic results using frequency domain signals after extracting fault characteristics.

The Type of Diagnostic Data	Rotating Speed	Diagnostic Accuracy
Frequency domain signal after extracting fault features	300 rpm	100%
Frequency domain signal after extracting fault features	500 rpm	100%
Frequency domain signal after extracting fault features	700 rpm	99%
Frequency domain signal after extracting fault features	900 rpm	100%

**Table 8 sensors-19-00591-t008:** Diagnostic results using frequency domain signals after extracting fault characteristics (Changing the parameters of DBN).

The Type of Diagnostic Data	Rotating Speed	Diagnostic Accuracy
Frequency domain signal after extracting fault features	300rpm	100%
Frequency domain signal after extracting fault features	500rpm	100%
Frequency domain signal after extracting fault features	700rpm	99%
Frequency domain signal after extracting fault features	900rpm	100%

**Table 9 sensors-19-00591-t009:** Diagnostic results using frequency domain signals after extracting fault characteristics (Changing the parameters of DBN).

The Type of Diagnostic Data	Rotating Speed	Diagnostic Accuracy
Frequency domain signal after extracting fault features	300rpm	100%
Frequency domain signal after extracting fault features	500rpm	100%
Frequency domain signal after extracting fault features	700rpm	99%
Frequency domain signal after extracting fault features	900rpm	100%

**Table 10 sensors-19-00591-t010:** The summary of results of each method based on neural networks.

Diagnosis Method	The Type of Diagnostic Data	Rotating Speed	Diagnostic Accuracy
Back propagation neural network	Symptom parameters	300 rpm	16%
Back propagation neural network	Symptom parameters	500 rpm	16%
Back propagation neural network	Symptom parameters	700 rpm	23%
Back propagation neural network	Symptom parameters	900 rpm	18%
Back propagation neural network	Principal component	300 rpm	17%
Back propagation neural network	Principal component	500 rpm	11%
Back propagation neural network	Principal component	700 rpm	16%
Back propagation neural network	Principal component	900 rpm	12%
Convolutional neural network	Symptom parameters	300 rpm	70%
Convolutional neural network	Symptom parameters	500 rpm	70%
Convolutional neural network	Symptom parameters	700 rpm	80%
Convolutional neural network	Symptom parameters	900 rpm	99%
Convolutional neural network	Principal component	300 rpm	70%
Convolutional neural network	Principal component	500 rpm	80%
Convolutional neural network	Principal component	700 rpm	80%
Convolutional neural network	Principal component	900 rpm	99%
Deep belief neural network	Time domain signal	300 rpm	20%
Deep belief neural network	Time domain signal	500 rpm	20%
Deep belief neural network	Time domain signal	700 rpm	20%
Deep belief neural network	Time domain signal	900 rpm	20%
Deep belief neural network	Frequency domain signal	300 rpm	80%
Deep belief neural network	Frequency domain signal	500 rpm	82%
Deep belief neural network	Frequency domain signal	700 rpm	98%
Deep belief neural network	Frequency domain signal	900 rpm	96%
Deep belief neural network	Frequency domain signal after extracting fault features	300 rpm	100%
Deep belief neural network	Frequency domain signal after extracting fault features	500 rpm	100%
Deep belief neural network	Frequency domain signal after extracting fault features	700 rpm	99%
Deep belief neural network	Frequency domain signal after extracting fault features	900 rpm	100%

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
