# Peer review of "A Precise Diagnosis Method of Structural Faults of Rotating Machinery based on Combination of Empirical Mode Decomposition, Sample Entropy, and Deep Belief Network"

_sensors, 2019, doi:10.3390/s19030591_

Round 1
Reviewer 1 Report
In this paper, the authors attempt to combine several techniques including the empirical mode decomposition, the sample entropy and the deep belief network to construct a diagnosis method for the identification of structural faults of rotating machinery. The proposed method has been applied to the real data from the rotation machinery. However, the major problem is that the paper has not clearly pointed out its major contributions in the abstract and the introduction section. In the abstract, the listed novelties are mostly summarization of the main results. Other comments are as follows.
The organization of the paper has not been presented at the end of section 1.
The paper lacks many technical details. For example, the last paragraph on page 221 is the training for the proposed model, not the training process for the Deep Belief Network (DBN) that the authors have mentioned.
The authors have used other deep learning models and presented the results in table 10. Why only mention DBN in the title?
Author Response
The responses to Reviewer 1:
Dear Reviewer 1:
Thank you very much for your careful review of our manuscript and kind suggestions. We have seriously and carefully revised the manuscript according to the comments from you and other reviewers. The revised contents in the manuscript are highlighted in red.
Your comments: In this paper, the authors attempt to combine several techniques including the empirical mode decomposition, the sample entropy and the deep belief network to construct a diagnosis method for the identification of structural faults of rotating machinery. The proposed method has been applied to the real data from the rotation machinery.
1. However, the major problem is that the paper has not clearly pointed out its major contributions in the abstract and the introduction section. In the abstract, the listed novelties are mostly summarization of the main results. Other comments are as follows.
Answer: The abstract has been rewritten in line 13.
The major contributions have been added to introduction in line 96.
2. The organization of the paper has not been presented at the end of section 1.
Answer: The organization of the paper has been added in line 118.
3. The paper lacks many technical details. For example, the last paragraph on page 221 is the training for the proposed model, not the training process for the Deep Belief Network (DBN) that the authors have mentioned.
Answer:
1. The description of the training process for the Deep Belief Network is added in line 227. And the new figure of the training process for the Deep Belief Network is added in line 237.
2. The details of experiment have been added in line 266.
4. The authors have used other deep learning models and presented the results in table 10. Why only mention DBN in the title?
Answer: The proposed diagnosis method focuses on deep belief neural network, so the diagnosis results of other deep learning models such as BPNN and CNN, are just applied to compare and verify the validity and accuracy of the proposed method. The summary of table 10 has been added in line 419.

Author Response
The responses to Reviewer 2:
Dear Reviewer 2:
Thank you very much for your careful review of our manuscript and kind suggestions. We have seriously and carefully revised the manuscript according to the comments from you and other reviewers. The revised contents in the manuscript are highlighted in red.
Your comments:
Hereby, I sending my comments to the paper: “A Precise Diagnosis Method of Structural Faults of Rotating Machinery based on Combination of Empirical Mode Decomposition and Sample Entropy and Deep Belief Network”.
1. In the title, there is an error, change and to “,” … Mode Decomposition, Sample Entropy and Deep Belief Network”.
Answer: The title has been modified and changed to: A Precise Diagnosis Method of Structural Faults of Rotating Machinery based on Combination of Empirical Mode Decomposition, Sample Entropy and Deep Belief Network.
2. In the abstract has to be included the obtained results, of the BPNN and CNN analysis.
Answer: The abstract has been rewritten and added the obtained results of the BPNN and CNN analysis.
3. Line 60, 91: It seems there a several places that the word abnormalities are using. My recommendation is to change to the word fault detection/machine faults.
Answer: In line 61 and line 92, the word “abnormalities” has been changed to the word “faults”.
4. Line 222 the text of figure 5 move to an earlier page.
Answer: In line 222, the text of figure 5 has been moved to the right place.
5. Line 248 change to: speeds of 300, 500, 700, and 900 rpms, respectively.
Answer: In line 266, the sentence has been changed to: speeds are 300, 500, 700, and 900 rpms, respectively.
6. 285 A part of the figure is cut.
Answer: In line 306, the original figure has been replaced by a new figure.
7. The quality of the photos can be increased.
Answer: The quality of the photos has been increased.
8. There is any section about the type of sensors and measurement system used in the experiments. It not addressed either the length of the samples and the sampling frequency used in the analysis.
Answer: The details have been added in line 266.
9. My major concerns are, first, the selection of the fault part of the signals and their respective harmonics are not documented. It has to be explain how it was obtained. The second one is the lack of a proper BPNN and CNN theory.
Answer: The details and figure of extracted fault signal reconstruction and the selection of the fault part of the signals are added in line 187.
This paper mainly shows the proposed precise diagnosis method based on extraction of fault information and DBN, and the method based on BPNN and CNN are used to compare and verify the effectiveness of the precise diagnosis method. The theory based on BPNN and CNN refers to document 31 and document 32, so the principle is not elaborated in detail herein.

Reviewer 3 Report
The approach suggested is a multi step approach with data filtering preceding the application of DBN.
The approach is demonstrated empirically without sensitivity analysis or scale up properties.
In addition, the DBN approach is purely diagnostic and does not provide prognostic and prescriptive capabilities, much needed in CBM.
English needs to be improved. For example:
l. 22 which proofed the validity of the proposed method.
Author Response
The responses to Reviewer 3:
Dear Reviewer 3:
Thank you very much for your careful review of our manuscript and kind suggestions. We have seriously and carefully revised the manuscript according to the comments from you and other reviewers. The revised contents in the manuscript are highlighted in red.
Your comments:The approach suggested is a multi-step approach with data filtering preceding the application of DBN.
1. The approach is demonstrated empirically without sensitivity analysis or scale up properties.
Answer: The main purpose of this paper is to diagnose structural faults of rotating machinery, and in particular to diagnose structural faults at low rotational speeds that were previously considered to be difficult to diagnose. The proposed algorithm properties have been validated through different settings and experiment data in Section 3.3 (line 311).
2. In addition, the DBN approach is purely diagnostic and does not provide prognostic and prescriptive capabilities, much needed in CBM.
Answer: As you pointed out, this paper mainly focuses the method of diagnosing the structural anomalies of rotating machinery, so we will make the analysis of prognostic and prescriptive capabilities as the content of continuing research in the future.
3. English needs to be improved. For example:
l. 22 which proofed the validity of the proposed method.
Answer: We have revised it and checked the English grammar.

Round 2
Reviewer 2 Report
Dear Authors:
Most of my recommendations have been adressed. In order to increase the quality of this paper I think its necessary some comments about what happen actually when the obtained presicion by neural networks is about 100%.
Author Response
The responses to Reviewer 2:
Dear Reviewer 2:
Thank you very much for your careful review of our manuscript and kind suggestions. We have seriously and carefully revised the manuscript according to the comments from you and other reviewers. The revised contents in the manuscript are highlighted in red.
Your comments:
Most of my recommendations have been adressed. In order to increase the quality of this paper I think its necessary some comments about what happen actually when the obtained presicion by neural networks is about 100%.
Answer: We have added and revised some presentation in line 345 and line 362.
After the decomposition of the original signal by EMD and the selection of intrinsic mode function with high signal-to-noise ratio based on the sample entropy, the selected intrinsic mode functions are reconstructed into new vibration signals. The number of samples based on the reconstructed signals for each state is 100, the length of each sample is 1000, 80% of the total number of samples are randomly sampled for training, and the rest of samples are used for the verification of trained model. There are four types of states in this experiment (normal state, misalignment state, unbalance state, and looseness of fasteners), so the total number of training samples is 320, and the total number of test samples is 80. The structure of the deep belief neural network is [1000, 100, 50, 4], the number of iterations of the sample is 100, the learning rate is 0.01. Based on the trained deep belief neural network, the diagnostic accuracy of the test samples is shown in table 7:
In the comparison experiment, the method of using the time domain signals only, the method of using frequency domain signals only and the proposed method are performed. The fault characteristics in the time domain signal is difficult to express, which will result in low diagnostic accuracy when only using time domain signals for diagnosis; When only using frequency domain signals for diagnosis, although the diagnostic accuracy is greatly improved compared with the method only using time domain signals, but the faults of the structural faults often have similar characteristics, the diagnostic accuracy is not enough satisfied. In this paper, the method of extraction of fault information based on empirical mode decomposition and sample entropy is proposed to screen and reconstruct the vibration signals with high SNR for precision diagnosis, and the frequency domain signals of reconstructed signals are combined with deep belief neural network which has strong classification performance. This method can increase diagnosis accuracy significantly and realize precise diagnosis of structural faults. This is because reconstructed vibration signals not only extract fault characteristics in original signals effectively, but also have extremely high SNR which is conducive to increase diagnosis accuracy greatly. The diagnosis accuracy was verified by changing parameters of DBN.
Reviewer 3 Report
revision is an improvement of initial submission. please apply suggestions of reviewers to the revision and submit final for publication.
Author Response
The responses to Reviewer 3:
Dear Reviewer 3:
Thank you very much for your careful review of our manuscript and kind suggestions. We have seriously and carefully revised the manuscript according to the comments from you and other reviewers. The revised contents in the manuscript are highlighted in red.
Your comments: revision is an improvement of initial submission. please apply suggestions of reviewers to the revision and submit final for publication.
Answer: Thank you for your kindly review our paper.